# Implementation and Cross-Validation of a Pharmacokinetic Model for Precision Dosing of Busulfan in Hematopoietic Stem Cell Transplanted Children

**DOI:** 10.3390/pharmaceutics14102107

**Published:** 2022-10-01

**Authors:** Sylvain Goutelle, Yann Thoma, Roxane Buffet, Michael Philippe, Thierry Buclin, Monia Guidi, Chantal Csajka

**Affiliations:** 1Hospices Civils de Lyon, Groupement Hospitalier Nord, Service de Pharmacie, 69005 Lyon, France; 2Univ Lyon, Université Lyon 1, UMR CNRS 5558, Laboratoire de Biométrie et Biologie Evolutive, 69622 Villeurbanne, France; 3Univ Lyon, Université Lyon 1, ISPB, Faculté de Pharmacie de Lyon, 69008 Lyon, France; 4School of Management and Engineering Vaud (HEIG-VD), University of Applied Sciences and Arts Western Switzerland (HES-SO), 1401 Yverdon-les-Bains, Switzerland; 5School of Pharmaceutical Sciences, University of Geneva, 1205 Geneva, Switzerland; 6Institute of Pharmaceutical Sciences of Western Switzerland, University of Geneva, 1205 Geneva, Switzerland; 7Institut d’Hématologie et d’Oncologie Pédiatrique & Centre Léon Bérard, Département de Pharmacie, 69008 Lyon, France; 8Service and Laboratory of Clinical Pharmacology, Department of Laboratory Medicine and Pathology, Lausanne University Hospital and University of Lausanne, 1011 Lausanne, Switzerland; 9Center for Research and Innovation in Clinical Pharmaceutical Sciences, Lausanne University Hospital and University of Lausanne, 1011 Lausanne, Switzerland

**Keywords:** busulfan, pediatrics, hematopoietic stem cell transplantation, oncology, population pharmacokinetics, therapeutic drug monitoring, model-informed precision dosing

## Abstract

Busulfan, a drug used in conditioning prior to hematopoietic stem cell transplantation (HSCT) in children, has a narrow therapeutic margin. The model-informed precision dosing (MIPD) of busulfan is desirable, but there is a lack of validated tools. The objective of this study was to implement and cross-validate a population pharmacokinetic (PK) model in the Tucuxi software for busulfan MIPD in HSCT children. A search of the literature was performed to identify candidate population PK models. The goodness of fit of three selected models was assessed in a dataset of 178 children by computing the mean error (ME) and root-mean-squared error of prediction (RMSE). The best model was implemented in Tucuxi. The individual predicted concentrations, the area under the concentration-time curve (AUC), and dosage requirements were compared between the Tucuxi model and a reference model available in the BestDose software in a subset of 61 children. The model from Paci et al. best fitted the data in the full dataset. In a subset of 61 patients, the predictive performance of Tucuxi and BestDose models was comparable with ME values of 6.4% and −2.5% and RMSE values of 11.4% and 13.6%, respectively. The agreement between the estimated AUC and the predicted dose was good, with 6.6% and 4.9% of the values being out of the 95% limits of agreement, respectively. To conclude, a PK model for busulfan MIPD was cross-validated and is now available in the Tucuxi software.

## 1. Introduction

Hematopoietic stem cell transplantation (HSCT) is a procedure used for treating various hematological malignancies (e.g., leukemia) as well as the severe forms of nonmalignant hematological diseases such as sickle-cell anemia, thalassemia, aplastic anemia, and severe combined immunodeficiencies [1]. Busulfan is an alkylating agent widely used in conditioning regimens prior to HSCT in both adults and children.

The PK of intravenous busulfan has been extensively reviewed elsewhere [2,3]. Briefly, busulfan binds at a low extent to both the plasma proteins and erythrocytes. Its average volume of distribution is large, comparable to that of the total body water (0.7 L/kg). The metabolism of busulfan is complex. It primarily involves conjugation to glutathione by the glutathione-S-transferase (GST) enzymes in body cells. Renal elimination is a minor elimination pathway, with 30% of a dose excreted into the urine, mostly as metabolites (about 1% is excreted as an unchanged drug). Its half-life is short, ranging from 1.5 to 3 h in most individuals.

Busulfan is usually administered over 4 days, as an intravenous infusion every 6, 12, or 24 h, depending on the HSCT center’s practice [4]. Busulfan has a narrow therapeutic margin. Its underexposure has been associated with graft rejection, while its overexposure has been associated with increased transplantation-related mortality [3,5,6]. An example of a potentially life-threatening complication of busulfan therapy is hepatic veno-occlusive disease.

The Busulfan area under the concentration-time curve (AUC) is the pharmacokinetic (PK) metric that best correlates with both the efficacy and toxicity of busulfan [5]. Busulfan’s AUC target is expressed as either the cumulative AUC (AUC_cum_) or the mean AUC per dose. The therapeutic drug monitoring (TDM) of busulfan has been recommended to determine drug exposure and adjust the drug dosage to avoid both under- and overexposure and to improve the outcome, especially in children [7,8,9]. The traditional method for estimating the AUC is based on rich sampling and noncompartmental analysis, but this method is laborious and uncomfortable for patients. An accurate estimation of busulfan’s AUC through sparse sampling is possible when combined with model-based analysis [10]. This is usually performed via dedicated PK software programs that perform Bayesian estimation of individual PK parameters. This Bayesian approach requires a population PK model that provides prior information for PK parameter estimation. A number of population PK models of busulfan in HSCT children have been published in the last two decades [3]. However, only a few models have been implemented in clinical PK programs and evaluated for drug dosing in an external manner. Neely et al. have developed and validated a nonparametric model for the Bayesian dose adjustment of busulfan in children, which is available in the BestDose^TM^ software [10]. An AUC estimation based on only two blood samples was shown to be accurate compared with the AUC estimated from rich PK profiles. In addition, the dosage adjustments based on this model in an external cohort of patients were consistent with the dosage adjustments calculated using noncompartmental analysis in a reference laboratory in the USA. This model can thus be considered the best standard for the model-guided TDM and dose adjustment of busulfan in children. BestDose is the latest version of a program formerly known as USC*Pack, MM-USC*Pack, and RightDose. It has been developed for more than four decades by the Laboratory of Applied Pharmacokinetics and Bioinformatics (LAPKB), at the University of Southern California, Los Angeles, USA. It is available from the laboratory website (http://www.lapk.org/ (accessed on 26 September 2022). More details are also available in a book edited by LAPKB directors [11]. However, BestDose only runs nonparametric PK models, which are scarce in population PK, as most published population models are parametric. Tucuxi is a new standalone program developed for the model-guided TDM of drugs [12]. It offers a user-friendly interface for clinicians, helping them in the process of drug dosage adjustment, by proposing concentration predictions, percentile calculations, and dosage adjustments. It has been developed by HEIG-VD in close collaboration with CHUV and is available via http://www.tucuxi.ch (accessed on 26 September 2022). The development team paid particular attention to user-friendliness and performance, to offer the best user experience to non-specialists. Its core computing engine is written in C++14, one of the best programming languages for efficient computation. Genericity is offered through the possibility to add new population PK models written in specific files, without modifying the software itself. Tucuxi only runs parametric PK models, but no busulfan model was available in the program so far. The objective of the present study was to implement and cross-validate a PK model for the AUC estimation and dosage adjustment of busulfan in HSCT children with the Tucuxi software, using BestDose as a reference.

## 2. Materials and Methods

### 2.1. Literature Search for Candidate Population PK Models of Busulfan

First, we searched the literature to identify candidate busulfan population PK models for implementation in Tucuxi. We conducted a PubMed search in July 2018, using the keywords “busulfan” and “population pharmacokinetics”. The search period was restricted to 10 years (2008–2018). We screened the references of retrieved papers as well. We examined the title and abstract of the identified articles and discarded some of them based on the following exclusion criteria: adult population only, nonparametric model, and model including gene polymorphism as covariates. The latter criterion was justified because the genetic data on glutathione-S-transferase, the main metabolic pathway of busulfan, were not available in our patients’ cohort. Indeed, this test is not performed in routine clinical practice in most centers.

As it was not feasible to implement and cross-validate all the available models, we performed a second round of selection based on the following exclusion criteria: part of the population with age >20 years, number of patients <20, model with the number of compartments ≥2, model not including body weight as a covariate, and reported without a description of all the necessary PK equations and parameters (e.g., fixed and random effects, residual error) for implementation in Tucuxi. The criteria related to the number of compartments and body weight covariate were justified by comparison with our reference model in BestDose.

### 2.2. Patients’ Data

We used a previously described dataset [13] to assess the predictive performance of the selected models and for subsequent analyses. Those data were collected as part of routine patient care in a hematology and oncology pediatric center in Lyon (Hospices Civils de Lyon, France), from June 2006 to June 2015. Ethics approval was obtained from two committees, including one from Hospices Civils de Lyon, as described previously [13]. We obtained a waiver of informed consent, as we only used the existing clinical data that were de-identified for this project.

Briefly, busulfan was administered over 4 consecutive days as a 2 h infusion administered every 6 h for 4 days (total of 16 doses) except in five patients (four were administered busulfan once daily and one twice daily). Busulfan TDM and dose individualization was performed for all the patients to achieve the conventional AUC target of 3.7–6.1 mg·h/L (900–1500 µM·min) per dose, corresponding to an AUC_cum_ of 60 to 100 mg·h/L. Two blood samples were obtained at 0.5 and 2 h after the end of the first busulfan infusion in each patient. For some patients, TDM was repeated with the same two-sample collection on the subsequent days of therapy on the physician’s demand. A validated high-performance liquid chromatography (HPLC) assay was used to measure the busulfan concentrations. More details about the drug assay are available elsewhere [10].

The data from 178 individuals (85 males and 93 females) were available, with 491 measured busulfan concentrations. The median (min, max) values of age, body weight, and height were as follows: 4.0 (0.13–21) years, 16.7 (3.2–90.9) kg, and 103 (46–190) cm, respectively.

### 2.3. Goodness of Fit of Selected Models

The three models selected after the literature search were imported into the NONMEM^®^ software. We created two versions of each model, with and without inter-occasion variability (IOV). Then, the models were used to fit the entire busulfan dataset by using the MAXEVAL = 0 option of NONMEM, which practically computes the individual Bayesian posterior parameter values based on both the prior information and the patients’ observations. We assessed the predictive performance of each model by computing the bias and imprecision of each individual posterior model’s predicted concentrations. The bias was defined as the mean percent error (*MPE*) of prediction:(1)$$MPE=\frac{1}{N}\sum_{i=1}^{N}\frac{Cpred_{i}-Cobs_{i}}{Cobs_{i}}$$
where *Cpred_i_* and *Cobs_i_* are individual predicted concentrations and observed concentrations, respectively.

The imprecision was defined as the root-mean-squared percent error (*RMSE)*:(2)$$RMSE=\sqrt{\frac{1}{N}\sum_{i=1}^{N}(\frac{Cpred_{i}-Cobs_{i}}{Cobs_{i}}){\;}^{2}}$$

The model providing the lowest values of bias and imprecision was considered the best model for our data, with imprecision being the primary criterion. The *MPE* and *RMSE* are classical criteria in the external validation of population PK models [14,15]. Diagnostic goodness-of-fit plots such as the individual predicted vs. observed concentrations, the conditional weighted residuals (*CWRES*) vs. population concentration predictions, and the time after dose further helped in assessing the most suitable model for our purpose.

### 2.4. Model Implementation in Tucuxi and Reference Model in BestDose

The model from Paci et al. [16] best fit the data in the NONMEM run (see the Results Section 3) and was imported into Tucuxi. This is a one-compartment model including the allometric scaling of busulfan clearance (*CL*) and the volume of distribution (*V*) to body weight (*BW*) as follows:(3)$$CL\left(L\cdot h^{-1}\right)=2.18\times {\left(\frac{BW}{9}\right)}^{\theta 1\;or\;\theta 2}$$
(4)$$V\left(L\right)=BW^{\theta 3}$$
where the *θ* parameters are the allometric power coefficients. For *CL*, the coefficient value depends on the patient’s weight, being *θ*1 = 1.25 in patients with *BW* < 9 kg and *θ*2 = 0.76 in patients with *BW* ≥ 9 kg; for *V*, it is *θ*3 = 0.86. The model without IOV was imported in Tucuxi because IOV is currently not handled by the program.

This model was fit to individual patients’ data in Tucuxi, and the results were compared with those obtained with our reference model, the nonparametric model from Neely et al. implemented in BestDose [10]. This is also a one-compartment model parameterized with an elimination rate constant (*Ke*) and *V*. The two parameters are allometrically scaled to *BW* or ideal body weight (*IBW*, estimated by the Traub–Johnson equation [17]), the lowest value being retained in each individual. The equations describing the covariate–parameter relationships are as follows:(5)$$Ke\left(h^{-1}\right)=Ke_{S}\times {\left(BW\;or\;IBW\right)}^{-0.25}$$
(6)$$V\left(L\right)=V_{S}\times \left(BW\;or\;IBW\right)$$

In Equations (5) and (6), *Ke_S_* and *V_S_* represent the “slope” parameters, which are random. Notably, the model published by Neely et al. also included age as a covariate on both *Ke* and *V*, with a complex polynomial relationship. This relationship could not be included in the desktop version of BestDose, so only BW/*IBW* was retained. It has been shown that this simpler model provided bias and imprecision as good as those of the full model [10].

Table 1 summarizes the parameter values of the models implemented in Tucuxi and BestDose. Notably, for the nonparametric model (BestDose), the mean and standard deviation do not fully represent the parameter distribution, and therefore, the median and minimal/maximal values are displayed. The typical values are reported for the parametric model.

### 2.5. Cross-Validation of Busulfan Model in Tucuxi

A probe subset of 61 individuals (*n* = 119 measured concentrations) who received busulfan every 6 h were randomly selected from the full dataset described above and used to compare the exposure and dosages predicted by the two models.

For each individual patient, the measured busulfan concentrations after the first dose were fitted with both the Tucuxi and BestDose models. The goodness of fit was assessed by computing the *MPE* and *RMSE* characterizing the comparison of observed concentrations and the individual posterior model predictions, as described previously. The AUC after the first dose (i.e., the first 6 h of therapy), denoted AUC_0–6_, was computed with both models, as well as the predicted future dose required to achieve a cumulative AUC set at 80 mg·h/L for the entire therapy. In this prediction, we assumed that no intra-individual (inter-occasion) variability in the PK parameters would occur, so those parameters estimated after the first dose would hold unchanged for the entire therapy of 16 doses over 4 days.

To assess the correlation and agreement between the concentrations, the AUC, and the predicted doses from the two models, we performed linear regression and Bland–Altman analyses, respectively.

## 3. Results

### 3.1. Literature Search and Analysis

Fifty-four reports were initially retrieved from PubMed. We discarded 38 studies based on the primary exclusion criteria. After this first round, 16 studies were kept, and 4 other studies were identified from citations in the literature. A total of 20 reports were examined in the second round, and 17 were excluded based on the secondary exclusion criteria. In the end, three models were retained for the goodness-of-fit analysis. Those were the models from Paci et al. [16], Booth et al. [18], and Trame et al. [19].

### 3.2. Goodness of Fit of the Three Candidate Models for the Entire Dataset

Table 2 summarizes the characteristics of the three models and their predictive performance in the analysis of the entire dataset after NONMEM runs, with the corresponding goodness-of-fit plots shown in Appendix A. For all the models, the predictive performance was improved when IOV was incorporated into the model. The model from Trame et al. showed the largest values of bias and imprecision. The models from Paci et al. and Booth et al. performed rather similarly regarding bias and imprecision. The model from Paci et al. was finally selected based on having the lowest imprecision and better *CWRES* diagnostic graphs than that of Booth et al. and then was imported into Tucuxi.

### 3.3. Comparative Analysis of Tucuxi and BestDose results

Table 3 shows the predictive performance of the models in Tucuxi and BestDose in the 61 probe patients (*n* = 119 measured concentrations). Compared with the reference model in BestDose, the test model in Tucuxi showed a small but significant bias (+6.4%), indicating a slight overprediction of concentrations, with somewhat lower imprecision. The BestDose model displayed indicated a smaller but still significant negative bias (−2.5%). The results from Tucuxi were highly correlated and concordant with the results from BestDose regarding the predicted concentrations (R^2^ = 0.93), the estimated AUC_0–6_ (R^2^ = 0.91), and the predicted doses to achieve the target AUC (R^2^ = 0.97), as shown in Figure 1.

Figure 2 represents the Bland–Altman analysis of the agreement between the results of the Tucuxi and BestDose models. For the predicted concentrations, the mean bias was 8.7 ± 12.7%, and 6 out of 119 values (5%) were out of the 95% limits of agreement. For the estimated AUC_0–6_, the mean bias was 3.7 ± 7.4%, and 4 out of 61 values (6.6%) were out of the 95% limits of agreement. For the predicted dose required to achieve the target cumulative AUC of 80 mg·h/L, the mean bias was −3.9 ± 8.1%, and 3 out of 61 values (4.9%) were out of the 95% limits of agreement. Overall, this analysis indicated a fairly good agreement between the exposures and dosage requirements estimated by the two models.

Regarding the outlier results, one patient (male, 15 years old, 39 kg, 153 cm) exhibited low observed busulfan concentrations at 0.5 and 2.5 h post-dose (730 and 140 ng/mL, respectively), the latter being especially low. The predicted concentrations from Tucuxi and BestDose at 0.5/2.5 h post-dose were 691/151 ng/mL and 430/230 ng/mL, respectively. Therefore, in this patient, Tucuxi fitted the data quite well, while BestDose underestimated the first observations and overestimated the second one. This resulted in large percent differences on the Bland–Altman plot (+46.6% and −41.5%). As a result, the estimated AUC_0–6_ was low, and the estimates were not consistent between the two programs either, with a 38.8% difference on the Bland–Altman plot. Notably, this patient had TDM again on the second day of therapy with the observed concentrations of 750 and 540 ng/mL at 0.5 and 2.5 h post-dose, which suggests that an error might have occurred for the second measurement on the first day. Another large discrepancy in the predicted concentrations (difference of +58.4% on the Bland–Altman plot) was observed in a very young patient (female, 2 months old, 3.2 kg, 45.5 cm) who exhibited a high concentration 2.5 h post-dose (630 ng/mL), just slightly lower than the concentration 0.5 h post-dose (800 ng/mL), which suggests a very low clearance, probably due to immature metabolism. Again, this observation was better fitted by the Tucuxi software. In this patient, the dosage requirements predicted by the two programs were not consistent (−41.2% on the Bland–Altman plot).

## 4. Discussion

Busulfan population PK in children has been examined in numerous reports. In their recent review, Lawson et al. identified 21 population PK models of busulfan in children published between 2007 and 2019 [3]. Among those, a few models were externally validated in terms of the predicted concentrations. However, very few studies reported the implementation of such models in Bayesian PK programs and their validation for dose prediction, which are key for model-informed prediction dosing (MIPD). To our knowledge, the model from Neely et al. implemented in BestDose [10] is the only model that has been externally validated in terms of both drug exposure and dosage requirements.

We aimed at implementing a parametric population PK model in the Tucuxi software and cross-validate its results with those from the reference model of Neely et al.

Among the three models selected from the literature, the model from Paci et al. [16] best fit the entire dataset in the NONMEM run. This model was developed in a large number of patients, including a significant proportion of very young patients (33% of the patients with BW < 9 kg). The median age and weight were the closest to those of our cohort. The model from Booth et al. also performed well. The model from Trame et al. appeared less adequate. Compared with our dataset, this model was developed in older children, and part of them (57%) received oral busulfan. Those characteristics may explain its poorer performance when applied to our data.

The model from Paci et al. was then imported into the Tucuxi software and was individually fitted to the data from a random subset of 61 probe patients. The results were compared with those of the reference model from Neely et al. The predicted concentrations from Tucuxi showed a slight positive bias, while those from BestDose showed a very slight negative bias. As a result, the difference between Tucuxi and BestDose predictions was positive on average (see Figure 2). However, the agreement appeared well acceptable clinically. Importantly, both the estimated AUC_0–6_ and the predicted doses were highly correlated and concordant between the Tucuxi model and the reference model.

Both scatter plots and Bland–Altman plots showed that the differences between the two programs were larger for the predicted concentrations compared with the AUC or doses. There are several reasons explaining this. First, there were twice more data on concentrations, compared with the AUC and doses, yet more variability. Second, the prediction errors in drug concentrations may compensate, e.g., the concentration at 0.5 h may be underestimated, and that at 2.5 h may be overestimated. Finally, the error propagation from the concentrations to the AUC and the predicted dose is not linear and depends on the sampling time. The AUC depends mostly on the estimation of the CL parameter. The concentration obtained 2.5 h after the dose is more informative about CL than that obtained at 0.5 h post-dose (this one being more informative about busulfan V). Therefore, a given difference in the concentrations measured 0.5 h post-dose may result in a lower difference in the CL, AUC, and dose estimates between the programs.

Overall, our results confirm the adequacy of the model implemented in Tucuxi and validate its performance for busulfan dosing in children. Both models appear to be practically interchangeable for the AUC estimation and dosing decision.

The discrepancies described in two patients in the Results section illustrate underlying the differences in the model implemented in BestDose and Tucuxi. BestDose uses nonparametric Bayesian estimation, while Tucuxi uses a classical maximum a posteriori (MAP) Bayesian estimation. More details about the parametric and nonparametric approaches in population PK and TDM programs are available in the previous publications from our groups [20,21,22]. With the nonparametric Bayesian approach, probabilities are updated when fitting the data, but the parameter values from the discrete prior distribution do not change. This might result in poor fit in some outlier subjects because no set of parameter values among those available in the prior is adequate. By contrast, the MAP approach is more flexible, because the prior and posterior distributions of PK parameters are continuous. However, this flexibility may also be a weakness in case of a sampling or measurement error, because the estimation may converge, and the apparent good fit will not warn users about the potential error. Strategies to cope with this, using, for example, the software-calculated percentiles to identify the out-of-scope measurements, are under investigation in Tucuxi. Notably, a hybrid fitting method not evaluated in the present study is available in BestDose and can better fit the outlier data [23].

Busulfan PK models are available in other dosing software tools, such as DoseMeRx, InsightRx, NextDose, and PrecisePK [24,25]. In a recent study, Lawson et al. compared the busulfan AUC predicted by models implemented in two different tools (NextDose and InsightRX) to the AUC calculated with the traditional trapezoidal rule, for various sampling strategies [26]. They showed that the bias and precision of the AUC estimation were not always consistent between software tools. However, they did not examine the agreement between the two PK programs. In addition, they only studied the AUC estimation, not the dosage requirements, which is arguably the most important criterion for MIPD. While the goodness of fit is a classic criterion in the validation of population PK models, we think that dose recommendations should be the primary diagnostic criterion when evaluating the dosing tools in a “fit-for-purpose” validation approach [27]. The previous studies evaluating dosing software considered this criterion as well [28,29].

This study has several limitations. The data were collected in routine patient care, and sampling for the drug’s concentration measurements was sparse. However, the reference model from Neely et al. was based on rich data and was externally validated based on part of this dataset [10]. The model from Paci et al. was also externally validated in the original study. Therefore, both models have been externally validated, and they are now cross-validated in the present study. We did not examine all the available models in our NONMEM analysis. Our primary goal was not to perform an external evaluation of all the published models but rather to identify a model to be cross-validated with our reference model implemented in BestDose. As our reference model was a one-compartment model, developed in children and adolescents only, including body weight and age as covariates, we preferred to select those models having similar characteristics to avoid confounders in the analysis of discrepancies. Model selection was based on both the goodness of fit and the characteristics facilitating the comparison with the reference model. Other models might be adequate as well and could be implemented in Tucuxi in the future. Indeed, it may be relevant to have several models available for a given drug in dosing software programs. This allows for a comparison of data fitting and dosage prediction for selecting the most adequate model for the population under consideration. It has been suggested that the best model should be chosen for each individual among a given set of competing models [28,30,31]. We did not evaluate the forecasting of concentration to be observed on subsequent TDM occasions, based on the first occasion data, because TDM was not repeated in all the patients. Although the models including IOV performed better in the NONMEM analysis, they could not be evaluated in the two dosing software programs, as neither currently handles IOV. By definition, IOV can be quantified a posteriori but is hardly predictable when designing a future dosage regimen. It seems that the best way to handle IOV in MIPD programs is to include IOV in the parameter estimation and data fitting but to ignore it in dose forecasting by considering IOV as an unexplained residual error [32,33]. Further research is necessary to confirm these findings.

To conclude, we implemented a PK model for busulfan dosing in children in the Tucuxi software. The model performance in terms of the estimated exposure and dosage prediction could be properly validated against a reference model available in BestDose. The results from both tools were consistent and indicated appropriate adequacy for the MIPD of busulfan in routine practice. Our approach may be proposed as a framework for the cross-validation of models used for MIPD in dosing software programs.

## Figures and Tables

**Figure 1 pharmaceutics-14-02107-f001:**
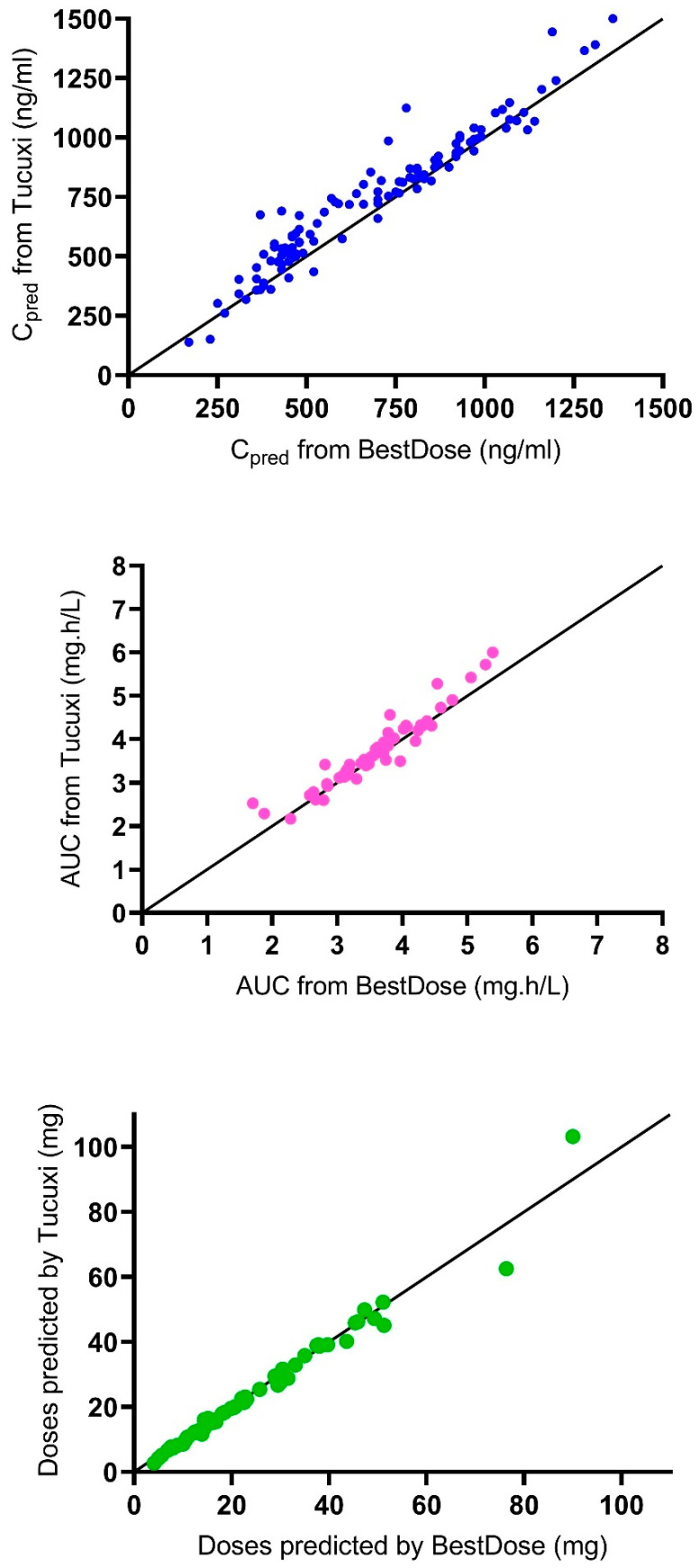
Correlation between results from Tucuxi and BestDose. Upper panel, predicted concentrations (C_pred_); medium panel, estimated AUC_0–6_; bottom panel, predicted doses. The solid line is the line of identity (y = x).

**Figure 2 pharmaceutics-14-02107-f002:**
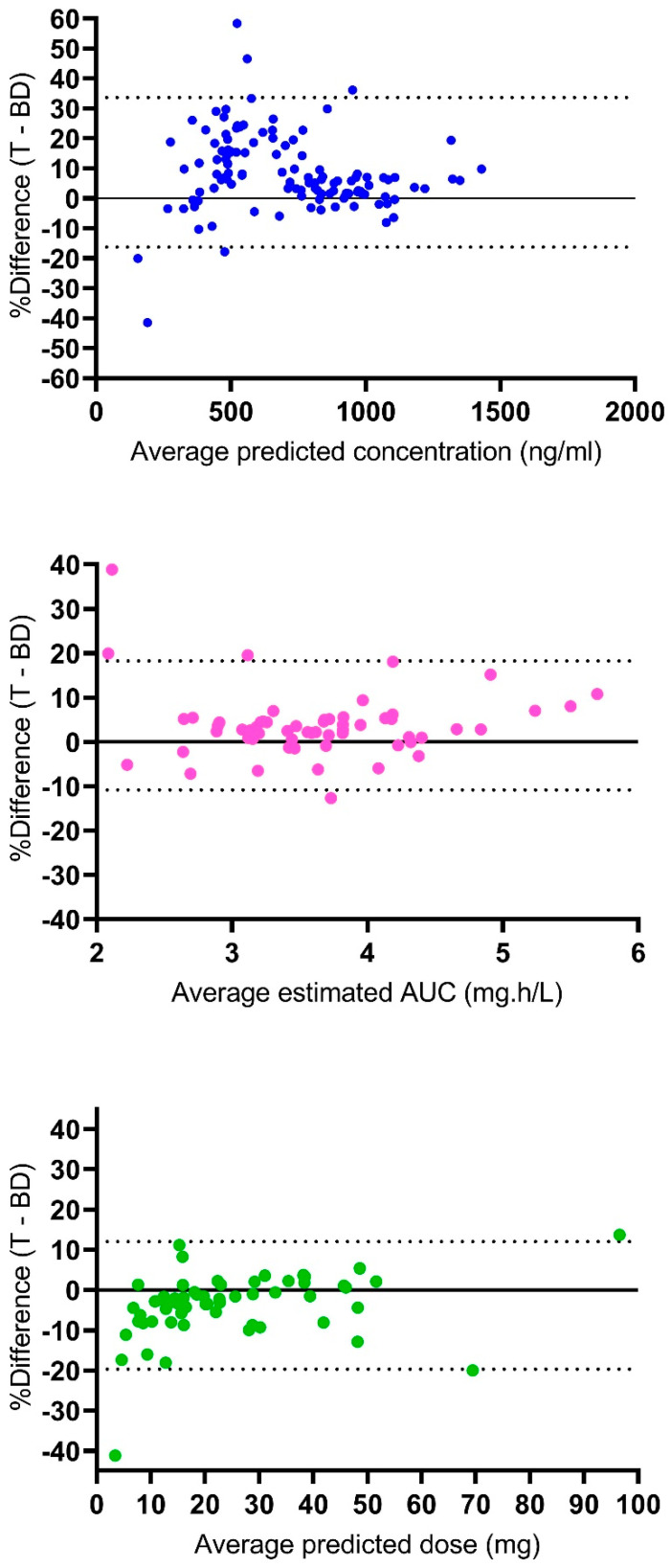
Bland–Altman plots of agreement between Tucuxi and BestDose results. Upper panel, agreement between predicted concentrations; medium panel, agreement between estimated AUC0–6; lower panel, agreement between predicted doses. The dotted lines are the 95% limits of agreement. The solid horizontal line is y = 0.

**Table 1 pharmaceutics-14-02107-t001:** Parameters of the busulfan PK models implemented in Tucuxi and BestDose.

Tucuxi Model [16]		BestDose Model [10]	
Parameter	Value	Parameter	Value
Typical CL (L/h)	2.18	Median Ke_S_ (h^−1^·kg^−0.25^) (min–max)	0.71 (0.42–0.98)
Typical V (L)	BW^0.86^	Median V_S_ (L·kg^−1^) (min–max)	0.72 (0.53–1.40)
CL variability (CV)	23%	Ke_S_ variability (CV)	18%
V variability (CV)	22%	V_S_ variability (CV)	21%
Residual error	11% (proportional) 57 ng/mL (additive SD)	Residual error ^a^	SD (mg/L) = 0.02 + 0.1·Cobs

^a^ In the residual error model, *Cobs* is measured busulfan concentration. Abbreviations: *BW*, body weight; *CL*, busulfan total body clearance; *CV*, coefficient of variation, *SD*, standard deviation.

**Table 2 pharmaceutics-14-02107-t002:** Main characteristics of the selected models and predictive performance in the entire dataset (*n* = 178 patients, 491 measured concentrations).

Model [Reference]	Population Characteristics	Model Characteristics	MPE (%) [95% Confidence Interval]		RMSE (%)	
			Model without IOV	Model with IOV	Model without IOV	Model with IOV
Paci et al. [16]	*n* = 205 Median age, 2.5 years (10 days–15 years) Median BW, 12 (3.5–62.5) kg	One-compartment CL~BW (allometric) V~BW (allometric) IOV on CL	2.7 [1.7–3.7]	1.6 [0.8–2.3]	12.2	8.7
Booth et al. [18]	*n* = 24 Mean age, 6.3 years (3 months–16.7 years) Mean BW, 23.8 (7.1–62.6) kg	One-compartment CL~BW (allometric) V~BW (allometric) IOV on CL and V	2.1 [1.0–3.3]	1.1 [0.3–1.9]	13.4	9.5
Trame et al. [19]	*n* = 94 Median age, 9.2 (0.4–18.8) years Median BW, 27.2 (4.2–80) kg	One-compartment CL~BW (allometric) V~BW (allometric) IOV on CL and V	7.1 [5.1–9.3]	5.0 [3.2–6.8]	26.2	22.2

Abbreviations: *BW*, body weight; *CL*, busulfan clearance; *IOV*, inter-occasion variability; *MPE*, mean prediction error; *RMSE*, root-mean-squared error; *V*, busulfan volume of distribution.

**Table 3 pharmaceutics-14-02107-t003:** Predictive performance of the test and reference model in the individual analysis of 61 patients (119 measured concentrations).

Model (Software)	MPE (%) [95% Confidence Interval]	RMSE (%)
Test model (Tucuxi) [16]	6.4 [4.7; 8.1]	11.4
Reference model (BestDose) [10]	−2.5 [−0.3; −4.7]	13.6

## Data Availability

Data are available upon request from the corresponding author.

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
