# Peer review of "Implementation and Cross-Validation of a Pharmacokinetic Model for Precision Dosing of Busulfan in Hematopoietic Stem Cell Transplanted Children"

_pharmaceutics, 2022, doi:10.3390/pharmaceutics14102107_

Round 1

Reviewer 1 Report

In this manuscript, the authors examined and cross-validated a population pharmacokinetic models in two softwares for busulfan MIPD in HSCT children.

The reviewer recognizes the manuscript is well-organized and contains valuable insight to the journal’s potential readers, however, several minor points should be revised before publication.

Major comment:

This is just only comment.

The reviewer strongly agrees that IOV terms should be included in the prediction software but also agrees the authors' view and comments they wrote as 'limitation'. All the three PPK models (Table 2) employed IOV and the model with IOV gave better RMSE than those without IOV, which means for busulfan PPK, IOV is quite important factor. The reviewer highly encourages and expects the authors to further study how to implement (or not) IOV effects in dose forcasting.

Minor comment

1) By comparing Cpred from BestDose and Tucuxi (Figure 1, upper, and Figure 2, upper), there seem to be larger bias than AUC and predicted Dose. Please discuss why this bias in Cpred was occured and less biases in AUC and Dose resulted.

2) Line 343, (see Figure 3) should be Figure '2'?

Author Response

Review Busulfan Tucuxi Pharmaceutics

Reviewer 1

In this manuscript, the authors examined and cross-validated a population pharmacokinetic models in two softwares for busulfan MIPD in HSCT children.

The reviewer recognizes the manuscript is well-organized and contains valuable insight to the journal’s potential readers, however, several minor points should be revised before publication.

 Major comment:

This is just only comment.

The reviewer strongly agrees that IOV terms should be included in the prediction software but also agrees the authors' view and comments they wrote as 'limitation'. All the three PPK models (Table 2) employed IOV and the model with IOV gave better RMSE than those without IOV, which means for busulfan PPK, IOV is quite important factor. The reviewer highly encourages and expects the authors to further study how to implement (or not) IOV effects in dose forcasting.

  1. Thank you for this comment. We fully agree with the importance of IOV and the need to take it into account in Bayesian forecasting.

Minor comment

1/ By comparing Cpred from BestDose and Tucuxi (Figure 1, upper, and Figure 2, upper), there seem to be larger bias than AUC and predicted Dose. Please discuss why this bias in Cpred was occured and less biases in AUC and Dose resulted.

  1. This is correct, both scatter plots and Bland-Altman plots show that the differences between the two programs were larger for Cpred compared with AUC or doses. We think that there are three main reasons for this. First, the amount of data was different between the three variables. There were twice more Cpred than AUC and predicted dose data, so there were more chances of larger differences. Second, for a given patient, prediction errors in drug concentrations may compensate, e.g., the concentration at 0.5h may be underestimated and that at 2.5h may be overestimated. As a result, the difference in terms of AUC and predicted dose can be lower. Finally, the error propagation from concentrations to AUC and predicted dose is not linear and depends on the sampling time. AUC depends mostly on the estimation of CL parameter. The concentration obtained 2.5h after the dose is more informative about CL than that obtained at 0.5h post-dose (this one being more informative about busulfan V). Therefore, a given difference in Cpred at 0.5h may result in a lower difference in CL, AUC and dose estimates.

A comment has been added in the discussion.

2/ Line 343, (see Figure 3) should be Figure '2'?

  1. Yes, thank you. Correction done.

Reviewer 2 Report

This manuscript describes the cross validation of Tucuxi and a model selected for implementation in this tool using BestDose and a pediatric dataset.

The manuscript is clearly written and the article presents an interesting approach, given that in the case of busulfan there is a good reference (BestDose).

Despite that there are some methodological problems that would need to be addressed.

Introduction

1.     Please add more details on the PK of busulfan to accurately present its ADME characteristics

2.     “Busulfan area under the concentration-time curve (AUC) is the pharmacokinetic (PK) index” please correct index to metric

Materials and Methods

3.     Despite I understand that the target population was children, The second round of model exclusion included the criteria part of population with age > 20 years, model with number of compartments ≥ 2, and model not including body weight as a covariate, is not sufficiently justified. A model that was developed with more patients is expected to be better even if the age range is larger, for example if a mode was developed with 30 patients age<20 and another with 200 patients with age range 1-60, probably the second model is going to be more accurate. Similarly, a model that contains over 2 compartments is not necessarily worse than a model that has 1 or 2. Please provide some rationale and/or a short description/table of the studies excluded due to these reasons.

4.     External model validation not be limited to biases (MPE) and imprecision (RMSE), graphical model validation should be done as well. Please add CWRES vs PRED and TIME, OBS VS PRED, OBS Vs IPRED, and VPC for each one of the models validated. If not enough space please add in the supplemental material.

5.     Please provide some more datails on the Tucuxi app, where was it developed? Who developed it ? What language is it using ? how is it working ? Where to find it ? A link maybe?

6.     Same for BestDose

Results

7.     From the results presented it cannot be concluded that the best model was used please add model validation plots as suggested above.

Discussion

8.     Please provide some more details regarding why BestDose was selected as reference to validate Tucuxi

Author Response

Review Busulfan Tucuxi Pharmaceutics

 Reviewer 2

This manuscript describes the cross validation of Tucuxi and a model selected for implementation in this tool using BestDose and a pediatric dataset.

The manuscript is clearly written and the article presents an interesting approach, given that in the case of busulfan there is a good reference (BestDose).

Despite that there are some methodological problems that would need to be addressed.

Introduction

1/ Please add more details on the PK of busulfan to accurately present its ADME characteristics

 Details about busulfan basic PK have been added in the introduction.

2/  “Busulfan area under the concentration-time curve (AUC) is the pharmacokinetic (PK) index” please correct index to metric

Changes done.

Materials and Methods

3/ Despite I understand that the target population was children, The second round of model exclusion included the criteria part of population with age > 20 years, model with number of compartments ≥ 2, and model not including body weight as a covariate, is not sufficiently justified. A model that was developed with more patients is expected to be better even if the age range is larger, for example if a mode was developed with 30 patients age<20 and another with 200 patients with age range 1-60, probably the second model is going to be more accurate. Similarly, a model that contains over 2 compartments is not necessarily worse than a model that has 1 or 2. Please provide some rationale and/or a short description/table of the studies excluded due to these reasons.

We understand the reviewers’ point, but our primary goal was not identify the best-fitting model, but to identify a model to be cross-validated with our reference model implemented in BestDose (Neely’s model) which has been thoroughly assessed for drug dosing. Because our reference model was a one-compartment model, developed in children and adolescents only, including body weight and age as covariates, we preferred to select models having similar characteristics to avoid confounders in the analysis of discrepancies. It has been shown that fitting data with a one- or a two-compartment model result in significant differences in the estimation of drug clearance, volume of distribution, and AUC (see PMID 9145878; https://doi.org/10.1093/jac/dkf168). Models developed in patients with large age range such as the model from McCune et al. (https://doi.org/10.1158/1078-0432.CCR-13-1960) include maturation covariates that were not available in our dataset.

We agree that other models could fit data well and be implemented in dosing software, but our goal was not to perform an external validation of all published model.

This has been clarified in the discussion about limitations.

4/ External model validation not be limited to biases (MPE) and imprecision (RMSE), graphical model validation should be done as well. Please add CWRES vs PRED and TIME, OBS VS PRED, OBS Vs IPRED, and VPC for each one of the models validated. If not enough space please add in the supplemental material.

Diagnostic graphs were added in the manuscript in the supplemental material as suggested by the reviewer. These graphs confirm the adequacy of the models to predict individual drug concentrations in our population, and the slightly better performances of the models of Paci et al and Booth et al compared to the model of Trame et al. In addition, the better CWRES distribution observed for the model of Paci et al compared to Booth et al further supported our model selection. Meanwhile, we decided not to show the VPCs in the manuscript as they are of little value to assess the goodness of individual model prediction.

5/Please provide some more datails on the Tucuxi app, where was it developed? Who developed it ? What language is it using ? how is it working ? Where to find it ? A link maybe ?

We added more information in the introduction, answering these questions. However, we chose not to go into too many details about how it works, as the reference 11 would allow the reader to find more details about the software implementation

6/Same for BestDose

More details about BestDose have been added in the introduction as well.

Results

7/From the results presented it cannot be concluded that the best model was used please add model validation plots as suggested above.

As explained above, our goal was not to identify the best-fitting model, but to identify and implement a model to be cross-validate with our reference model. This has been clarified in the discussion.

Discussion

8/Please provide some more details regarding why BestDose was selected as reference to validate Tucuxi

These details are available in the introduction. “Neely et al. have developed and validated a nonparametric model for Bayesian dose adjustment of busulfan in children, which is available in the BestDoseTM software [10]. AUC estimation based on only two blood samples was shown to be accurate compared with AUC estimated from rich PK profiles. In addition, dosage adjustments based on this model in an external cohort of patients were consistent with dosage adjustments calculated by non-compartmental analysis in a reference laboratory in the USA. This model can thus be considered as the best standard for model-guided TDM and dose adjustment of busulfan in children.”

This is also mentioned in the discussion: “To our knowledge, the model from Neely et al. implemented in BestDose [10] is the only model that has been externally validated in terms of both drug exposure and dosage requirements.”

Round 2

Reviewer 2 Report

The manuscript is significantly improved and there are no further comments.